# A secreted antibacterial neuropeptide shapes the microbiome of *Hydra*

René Augustin[1], Katja Schröder[1], Andrea P. Murillo Rincón[1], Sebastian Fraune[1], Friederike Anton-Erxleben[1], Eva-Maria Herbst[1], Jörg Wittlieb[1], Martin Schwentner[1,4], Joachim Grötzinger[2], Trudy M. Wassenaar[3] & Thomas C.G. Bosch[1]

Colonization of body epithelial surfaces with a highly specific microbial community is a fundamental feature of all animals, yet the underlying mechanisms by which these communities are selected and maintained are not well understood. Here, we show that sensory and ganglion neurons in the ectodermal epithelium of the model organism hydra (a member of the animal phylum Cnidaria) secrete neuropeptides with antibacterial activity that may shape the microbiome on the body surface. In particular, a specific neuropeptide, which we call NDA-1, contributes to the reduction of Gram-positive bacteria during early development and thus to a spatial distribution of the main colonizer, the Gram-negative *Curvibacter* sp., along the body axis. Our findings warrant further research to test whether neuropeptides secreted by nerve cells contribute to the spatial structure of microbial communities in other organisms.

[1] Zoological Institute and Interdisciplinary Research Center Kiel Life Science, University of Kiel, 24098 Kiel, Germany. [2] Institute of Biochemistry, University of Kiel, 24098 Kiel, Germany. [3] Molecular Microbiology and Genomics Consultancy, 55576 Zotzenheim, Germany. [4]Present address: Museum of Comparative Zoology, Harvard University, Cambridge, MA 02138, USA. René Augustin and Katja Schröder contributed equally to this work. Correspondence and requests for materials should be addressed to T.C.G.B. (email: tbosch@zoologie.uni-kiel.de)

There is an increasing appreciation that homeostasis of any organism depends on a constant dialogue with the microorganisms that cover its surfaces[1–3] and this also applies to aquatic animals[4, 5]. Each organism maintains a specific microbiome, comprised of a stable core as well as variable components. However, how exactly the formation of a multi-level species microbiome is regulated, how the resultant holobiont operates as a functional unit, and by which mechanisms the host interacts with its microbiome, remains largely unknown.

Like all living organisms, animals are constantly being exposed to microbes; this applies to their external (skin, exoskeleton) as well as internal (respiratory, gastrointestinal) surfaces. Biologically active peptides provide bidirectional interactions between such host tissues and the microbiome. The body compartments of animals are innervated by a dense network of nerve cells that produce neuropeptides, which serve as messengers in the complex interactions within and between nerve cells and the connected body parts[6]. The microbiome interacts with nerve cell endings in surface tissues, for instance by inducing pain[7], while host-derived neuropeptides in turn have been proposed to interact with the microbiome[6–13]. However, the impact of neuropeptides on the composition of a host-specific microbiome has not been studied in detail.

Here, we tested the hypothesis that neuropeptides may be involved in the interaction and communication between the host and its natural array of microbes. For this we used the model organism hydra, a member of the animal phylum Cnidaria, to which corals, jellyfishes, and polyps belong. Cnidaria is the sister clade of Bilateria and among the first metazoans that contain neurons. Cnidarian nervous systems function as diffuse nerve nets of approximately 3000 neurons and offer great potential for understanding the basic design principles of a nervous system[14].

Hydra is covered by a microbiome that emerges progressively after hatching via a conserved temporal pattern, to reach a stable resident microbiome at adult stage[15]. The latter is characterized by Gram-negative bacterial species with a preponderance of *Curvibacter sp.*, a member of the β-Proteobacteria[15]. Disturbances or shifts in any of these bacterial colonizers can compromise the health of the whole animal[16].

In a previous study, we had observed that the absence of neurons affects the associated bacterial community and leads to an increase of Bacteroidetes and reduced levels of β-Proteobacteria, suggestive of neuron-mediated selective forces on the associated microbiota[17]. Here, we investigated whether such effects are due to neuropeptides, which we hypothesized to serve an antimicrobial, innate immune function that shapes the hydra's microbiome.

## Results

**Neuron development in Hydra coincides with decreasing Gram-positives.** The formation of a functional nervous system during the development of hydra polyps from a hatching egg via juvenile to the adult polyp stage was quantified microscopically. Based on previous observations in *Hydra vulgaris* strain AEP[18], very few nerve cells (0.02 neurons per epithelial cell) are present in hatchlings. By 3 weeks a complete nervous system is present, with approximately 0.25 neurons per epithelial cell (Fig. 1a). The composition of the bacterial communities associated with hydra during ontogeny has been assessed previously by culture-independent pyrosequencing of bacterial DNA[15]. This resulted in the finding of dramatic changes in bacterial composition over time and a stable microbiome was observed 3 to 4 weeks after hatching[15]. Reanalysis of that data showed that the ratio of Gram-positive bacteria decreased approximately threefold within the first 2 weeks after hatching and continued to decrease, to comprise <1% of the microbial community at week 15 (Fig. 1b). This was accompanied by the establishment of a consistent bacterial composition dominated by *Curvibacter* sp.

The high variation in abundance of Gram-positive bacteria between week 3 and 6 (Fig. 1b) was shown previously[15] to be a characteristic feature of the maturation of the adult microbiome in hydra.

**Locally expressed neuropeptide NDA-1 functions as an antimicrobial.** When screening for hydra peptides with structural similarity to invertebrate cytokines, so-called astakines[19], we noticed a peptide, which we call NDA-1, that contains a signal sequence followed by a cationic 71 aa-long peptide containing ten conserved cysteine residues (Fig. 2a). A neighbor-joining phylogenetic tree was constructed but remained largely unresolved with regard to the relationship of the included prokineticins, astakines, dickkopf, and colipase gene families: apart from the cysteine pattern no congruence was observed between NDA-1 and the other peptides (Supplementary Fig. 1). An NDA-1 gene is present in the *Hydra magnipapillata* genome and in the transcriptome of three other hydra species including *H. vulgaris* AEP[20]. Since there are no identifiable orthologs to NDA-1 outside hydra, it is currently considered to represent a taxon-restricted gene (Supplementary Fig. 1).

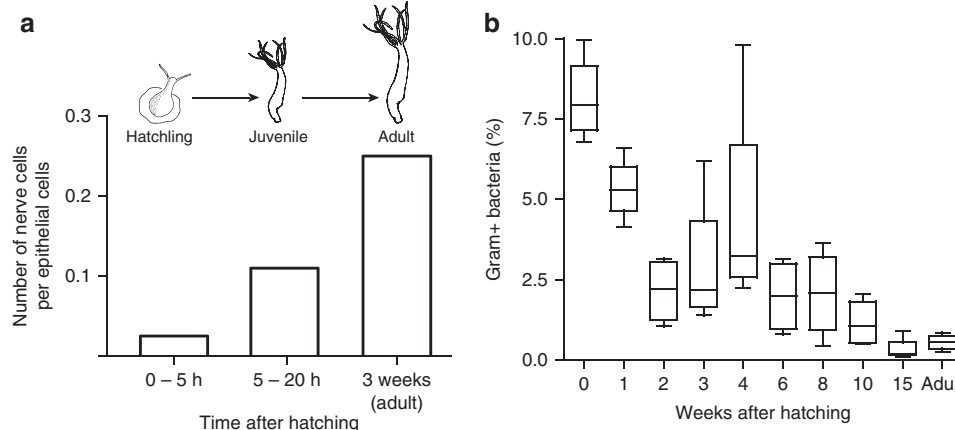

**Fig. 1** Formation of a neural system and a microbiome during hydra development. **a** Quantitative assessment of the nervous system in three developmental stages; original data taken from ref. [18]. **b** Proportion of Gram-positive bacteria in the microbiome over time plotted as *box*- and *whisker*-plot with the *box* containing 25th to 75th percentiles, the *line* representing the *mean* and *whiskers* showing maximum and minimum values; original data taken from ref. [15]

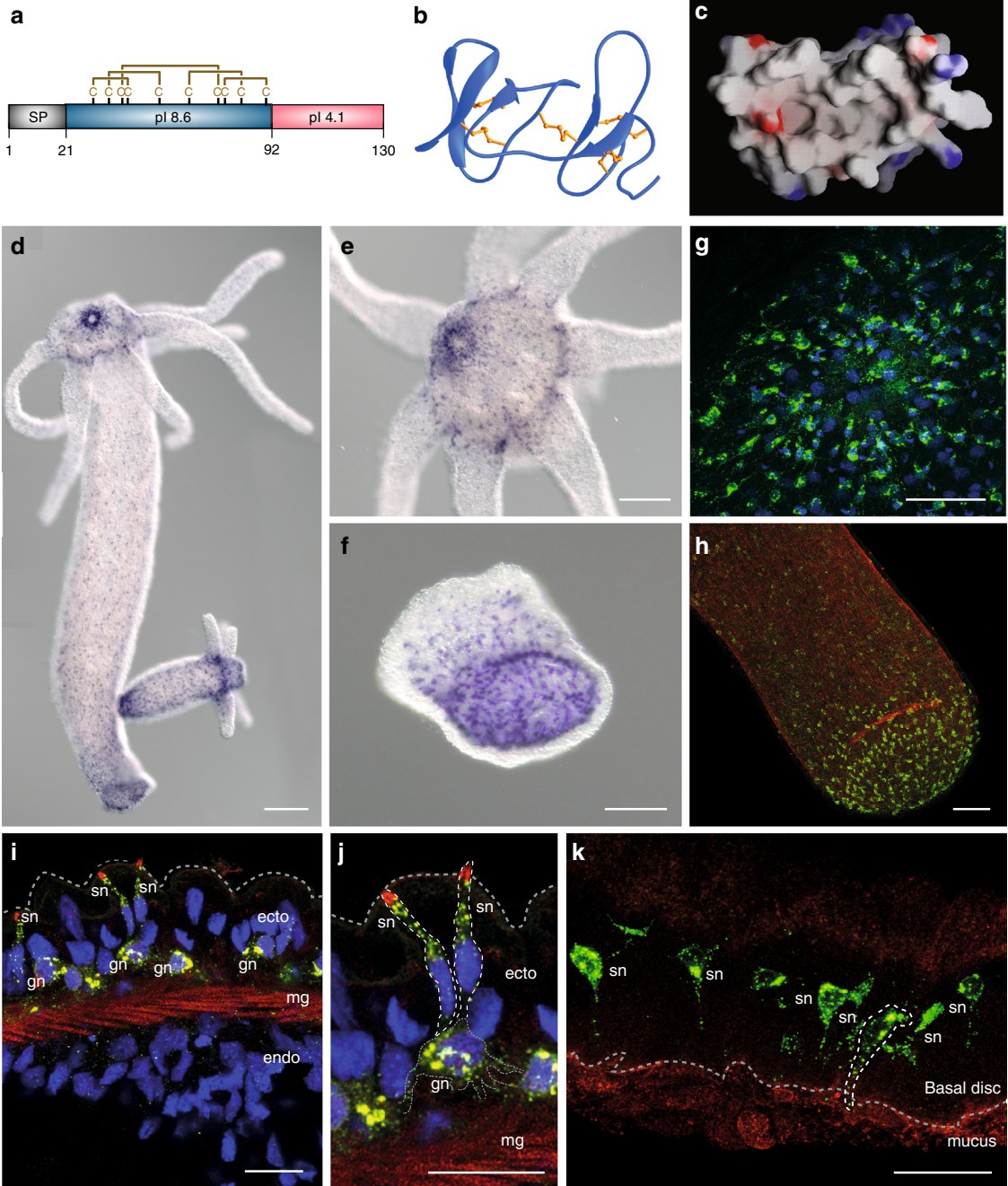

**Fig. 2** Biochemical domains and expression of neuropeptide NDA-1. **a** Schematic representation of NDA-1 precursor peptide showing its signal peptide (*SP*), cysteine-rich cationic domain (*blue*) with predicted disulfide bonds indicated, and C-terminal domain (*red*). **b** Predicted tertiary structure of mature NDA-1 peptide. **c** Molecular surface presentation showing a hydrophobic pocket with localized anionic (*red*) and cationic (*blue*) residues. **d–f** In situ hybridization demonstrating gene expression of NDA-1 mRNA in the neurons of **d** an adult polyp, with high expression in **e** the head and **f** the foot. *Scale bar* **d**: 200 μm, **e**, **f**: 100 μm. **g–k** Immuno-cytochemistry of NDA-1 peptide (*green*) combined with TO-PRO3-stained cellular DNA (*blue*) and (except in **k**) rhodamin-phalloidin for F-actin (*red*). Dense population of NDA-1 positive neurons are located in **g** the head and **h** the foot. *Scale bar*: 50 μm. **i–k** Confocal laser scanning microscope showing intracellular location of NDA-1 peptide in **i**, **j** the hypostomal region of the head, where ganglion neurons (gn) and sensory neurons (sn) are located in the ectodermal (ecto) layer; mesoglea (mg), and endoderm (endo) are indicated. *Scale bar*: 16 μm. **k** Confocal laser scanning microscope showing intracellular location of NDA-1 peptide in sensory neurons of the foot, with 1A10-antibody detecting an ectodermal surface epitope (*red*). A sensory neuron expressing NDA-1 is marked by a *broken white line*. *Scale bar*: 20 μm

The predicted structure of the cationic domain of NDA-1 is shown in Fig. 2b. The signal peptide is followed by a segment containing two β-sheets that are separated by a long flexible loop and are held together by five disulfide bonds with similarity to a colipase fold. Molecular surface prediction indicates the formation of a strongly hydrophobic pocket (Fig. 2c). Taken together, the predicted structure, its charge distribution and the cysteine pattern all support the view that NDA-1 is produced as a precursor: cleavage of the 38 aa C-terminal would produce a mature peptide of 71 aa.

By means of in situ hybridization we determined that the gene for NDA-1 was expressed in neurons throughout the body

(Fig. 2d–f). High numbers of NDA-1 gene-expressing ganglion and sensory neurons are localized in the head (hypostome) and foot of the polyp, whereas in the body column fewer ganglion cells express the NDA-1 gene. To localize the peptide in hydra tissue a polyclonal antiserum was produced, which detected mature peptide in ganglion as well as sensory neurons interspersed in the ectodermal epithelial layer (Fig. 2g, h). Confocal laser scanning microscopy revealed that NDA-1 was located in the distal parts of the sensory neurons, facing the outer mucus layer (Fig. 2i, j). Similarly, in sensory neurons in the foot the peptide was detected in the protrusions reaching out to the ectodermal surface (Fig. 2k), suggesting that the peptide is secreted into the mucus layer.

Since the mucus layer is the habitat of most of the resident microbiome[16, 21], we next assessed whether the mature peptide had antimicrobial activity. For this, recombinant mature peptide (rNDA-1) was produced in *E. coli*. After partial purification, microdilution susceptibility assays were performed to determine the minimal inhibitory concentration (MIC) of rNDA-1 against a range of Gram-negative and Gram-positive bacteria. As summarized in Table 1, the peptide was highly toxic for the Gram-positive freshwater species *Bacillus megaterium*, *Trichococcus pasteurii*, and *Trichococcus collinsii*, whose growth was inhibited by concentrations below 1 µM. Growth of *B. megaterium* was affected at a concentration of 400 nM. In contrast, a strain of *Pseudomonas* sp. that had been isolated from a biofilm in hydra

**Table 1 Minimal inhibitory concentration values of NDA-1 against a selection of Gram-positive and Gram-negative bacteria**

|  | Bacterial strain | Habitat | NDA-1 MIC (µM) |
|---|---|---|---|
| Gram-positives | *Bacillus megaterium* ATCC 14581 | Soil, fresh water | 0.4 |
|  | *Staphylococcus aureus* ATCC 12600 | Human skin, mucosal surfaces | 23.1 |
|  | *Bacillus subtilis* DSM 10 | Soil, fresh water | 5.8 |
|  | *Trichococcus pasteurii* DSM 2381 | Fresh water | 0.9 |
|  | *Trichococcus collinsii* DSM 14526 | Fresh water | 0.4–0.9 |
| Gram-negatives | *Escherichia coli* K12 MG1665 | — | >14.0 |
|  | *Curvibacter* sp. | *H. vulgaris* AEP main colonizer | 0.4 |
|  | *Acinetobacter* sp. (Biofilm 4) | *H. vulgaris* AEP culture dish | 7.0 |
|  | *Pseudomonas* sp. (Biofilm 3) | *H. vulgaris* AEP culture dish | >20.9 |

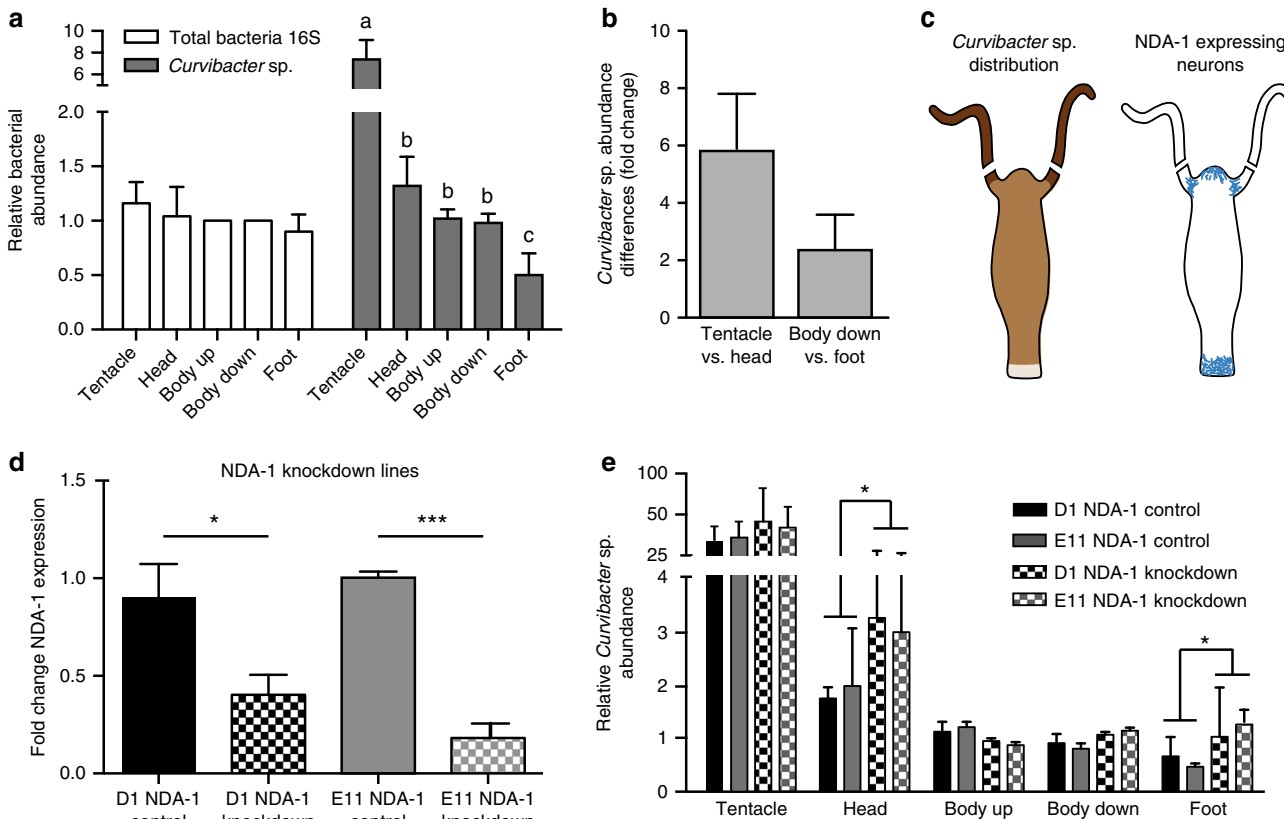

**Fig. 3** Spatial distribution of *Curvibacter* is regulated by NDA-1. **a** Abundance of total bacterial loads (*white bars*) and *Curvibacter* sp. (*gray bars*) at various body parts, with standard errors of the mean. Statistical differences (*P* < 0,001, GLM; *n* = 5) are indicated by different letters. **b** Relative abundance of *Curvibacter* sp. on different body parts. **c** Schematic representation of the quantitative distribution of *Curvibacter* (*left*) and NDA-1-expressing sensory neurons (*right*). **d** NDA-1 expression in hydra knockdown lines D1 and E11 with their corresponding controls; statistical significance: *\*P* < 0,05, *\*\*\*P* < 0,001 ANOVA; *n* = 3. **e** Body site distribution of *Curvibacter* sp. in NDA-1 knockdown lines and controls (GLM; *n* = 3)

culture dish was not inhibited, though significant activity was observed against a likewise isolated biofilm-forming *Acinetobacter* sp. Interestingly, NDA-1 was particularly potent against *Curvibacter* sp., the main colonizer that can reach over 70% abundance in the hydra microbiome[16]. We speculate that the observed antimicrobial activity of the peptide is due to interactions of its hydrophobic pocket with bacterial membranes. Based on the observed expression pattern and antimicrobial activity, we propose that NDA-1 may regulate the composition of the hydra microbiome, both qualitatively (regulating the ratio between Gram-negative and Gram-positive bacteria) and quantitatively, by keeping the numbers of bacteria in check.

**NDA-1 determines the spatial distribution of *Curvibacter* sp**. The high antimicrobial activity against *Curvibacter* sp. raised the question about consequences with respect to the local distribution of this main colonizer along the body. Total bacterial load and *Curvibacter* abundance were determined for various body parts. While total bacterial load was found to be evenly distributed along the body column (Fig. 3a), *Curvibacter* had a higher mean abundance in the tentacles than in other body regions (Fig. 3a–c). An additional difference was observed between the lower body column and the foot tissue with very low abundance of *Curvibacter* in foot tissue (Fig. 3a–c). In head tissue, where NDA-1 is strongly expressed by sensory neurons around the hypostome and in the base of tentacles (as shown in Fig. 2e), a sixfold reduction in *Curvibacter* abundance is observed compared to tentacles (Fig. 3b). Conversely, when comparing body column and foot tissue, *Curvibacter* abundance is twofold lower in foot tissue, which has a high density of NDA-1-expressing and secreting neurons (Fig. 3b, c).

To elucidate whether there is a direct relationship between presence of NDA-1 secreting neurons and the drastic change in *Curvibacter* abundance in tentacle vs. head and body column vs.

foot region (Fig. 3b), NDA-1 expression was diminished by means of antisense technology. Two transgenic lines of hydra were established that produced a 60% (line D1) and 80% (line E11) decrease of the endogenous NDA-1 transcript, as determined by quantitative real-time PCR (Fig. 3d). Control lines originating from the same mosaic founder polyp lacking eGFP-positive cells were used to measure the effect of the transgenic interference within a conserved genomic background. Supplementary Fig. 2 shows confocal laser scanning micrographs of the foot region, illustrating reduced expression of NDA-1 in the knockdown lines. The NDA-1 knockdown animals showed no differences in mechanical-induced contractile behavior and feeding response when compared to the corresponding control lines, indicating that NDA-1 knockdown does not appear to have functional consequences regarding common behavioral responses in Hydra (Supplementary Figs. 3 and 4). As expected, the decreased expression of NDA-1 in both transgenic lines affected the spatial distribution of *Curvibacter* (Fig. 3d), resulting in a higher *Curvibacter* abundance in the head and foot compared to control lines. This suggests that neuropeptide NDA-1 in wild-type polyps contributes to the reduction of *Curvibacter* abundance of head and body compared to tentacle tissue, and prevents *Curvibacter* growth at the very basal end of foot tissue. The neuropeptide, therefore, appears to contribute to the spatial distribution of hydra's main colonizing bacteria along the body column.

Interestingly, attempts to overexpress NDA-1 ectopically in epithelial cells resulted in a lethal phenotype early after hatching, suggesting that NDA-1 may have not yet discovered functions in addition to its role in host–microbe interaction (Supplementary Fig. 5).

**Other hydra neuropeptides also have antimicrobial activity**. Previous works had already identified a large number of peptides

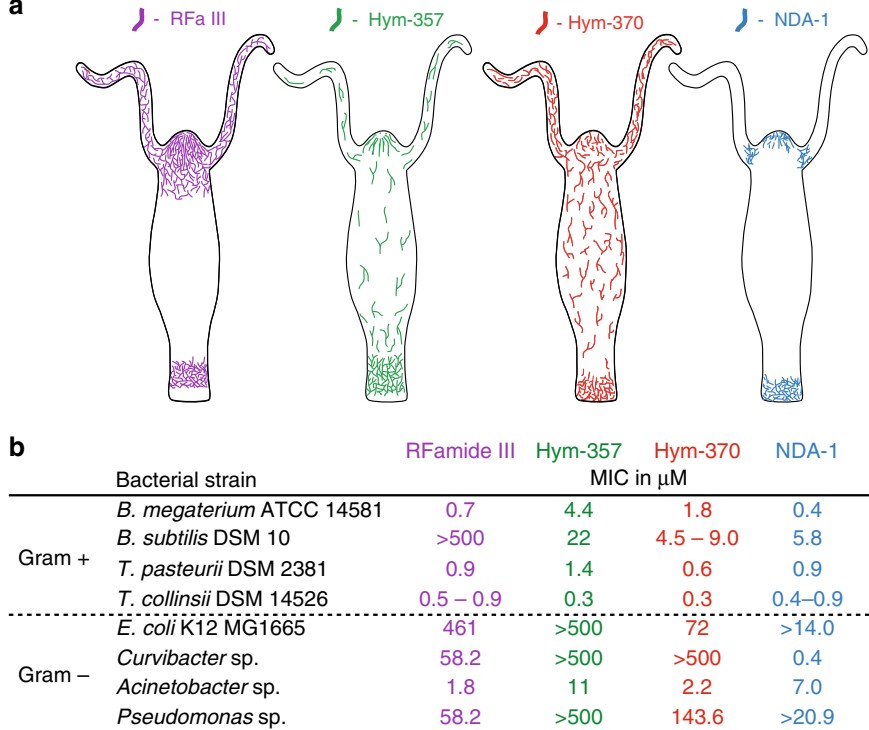

**Fig. 4** Local distribution of expression and antibacterial activity of three neuropeptides. **a** Schematic drawing of neurons expressing RFamide III (RFa III), Hym-357, and Hym-370 along the body of hydra. **b** Antibacterial activity of the three neuropeptides. Data for Hym-357 taken from ref. [22], for Hym-370 from ref. [23], for RFamide III from refs [48, 49]

| | Bacterial strain | RFamide III | Hym-357 | Hym-370 | NDA-1 |
|---|---|---|---|---|---|
| | | | MIC in µM | | |
| Gram + | *B. megaterium* ATCC 14581 | 0.7 | 4.4 | 1.8 | 0.4 |
| | *B. subtilis* DSM 10 | >500 | 22 | 4.5 – 9.0 | 5.8 |
| | *T. pasteurii* DSM 2381 | 0.9 | 1.4 | 0.6 | 0.9 |
| | *T. collinsii* DSM 14526 | 0.5 – 0.9 | 0.3 | 0.3 | 0.4–0.9 |
| Gram − | *E. coli* K12 MG1665 | 461 | >500 | 72 | >14.0 |
| | *Curvibacter* sp. | 58.2 | >500 | >500 | 0.4 |
| | *Acinetobacter* sp. | 1.8 | 11 | 2.2 | 7.0 |
| | *Pseudomonas* sp. | 58.2 | >500 | 143.6 | >20.9 |

characterized as neurotransmitters or neuromodulators in hydra, although NDA-1 was not among them. In line with the discovery of NDA-1 functioning as an antimicrobial peptide, we assessed whether other, pre-identified neuropeptides could also have an impact on the bacterial microbiome. Predicted biologically active forms of hydra neuropeptides were screened in silico for potential antimicrobial activity, based on their biochemical parameters such as charge distribution, isoelectric point and amphipathicity. Three neuropeptides, representing the three major neuropeptide classes of hydra were selected: Hym-370, Hym-357, and RFamide III. Hym-370 and Hym-357 had been identified in a screen for myoactive peptides[22–25]. Rfamide III is a neuropeptide belonging to a family of FMRFamide-like peptides conserved in invertebrates and vertebrates[26–29]. Previous publications[26–30] have demonstrated that the expression patterns of these three genes are unique in that each gene is expressed in different subpopulations of neurons, forming distinct localized expression patterns along the body axis with sharp boundaries (Fig. 4a). To assess their antimicrobial activity, the neuropeptides were chemically synthesized and tested in microdilution susceptibility assays against eight bacterial species. The results showed that all three peptides had antibacterial activity (Fig. 4b). Hym-357 and Hym-370 were active against all four tested Gram-positive bacteria (*B. megaterium*, *B. subtilis*, *T. pasteurii*, and *T. collinsii*), but RFamide III was inactive against *B. subtilis*. Variation in antimicrobial activity was also observed for the four Gram-negative bacteria tested, whereby not one of the tested bacteria species was insensitive to all three (Fig. 4b). In contrast to NDA-1, none of the three peptides was active against the main colonizer *Curvibacter* sp. These results suggest that distinct nerve cells may contribute to the spatial structure of hydra's microbial community by expressing a variety of neuropeptides with specific antimicrobial activity. In contrast to the antimicrobial peptides identified in hydra so far[31–34], the neuropeptides analyzed here are active mostly against Gram-positive bacteria. Additionally, the main Gram-negative colonizer *Curvibacter* is differentially affected by the peptides.

## Discussion

Here, we have shown that the composition of the microbiome in hydra, a member of the ancient animal phylum Cnidaria, is regulated by NDA-1, a neuron-secreted peptide, and possibly by other neuropeptides. Using in vivo transgenesis and in vitro antimicrobial activity assays, we have demonstrated that NDA-1 affects the hydra microbiome by inhibiting the growth of Gram-positive bacteria and of the main colonizer, the Gram-negative bacterium *Curvibacter* sp. In addition, we identified a number of previously characterized neuropeptides to have specific antimicrobial activity, adding to previous reports showing that some neuropeptides from other organisms can display direct antibacterial activity[8–13].

Microbial colonization in early hydra embryos is controlled by maternally encoded antimicrobial peptides[35]. After mid-blastula transition, zygotically expressed antimicrobial peptides take control of the microbiome. After hatching, a stable microbiome is established within 3 to 4 weeks[15]. Symbiotic microbes may be acquired by both vertical and horizontal transmission[35]. In adult hydra, stably associated microbes play a role in pathogen defense[16].

Due to similarities in amino-acid composition, amphipathic design, cationic charge and size, it has been suggested that many neuropeptides and peptide hormones may be directly involved in innate immune reactions against pathogenic intruders[36]. In mammals, neuropeptides such as substance P[37], calcitonin gene-related peptide, neuropeptide Y, vasoactive intestinal polypeptide,

somatostatin and corticotropin-releasing factor, have all been suggested as participating in the bidirectional gut-brain communication[6]. However, their impact on the composition of the microbiota, or their role in interaction between this and the nervous system, are still to be assessed. Our data reveal that several neuropeptides in hydra may play roles in the bidirectional relationship between the host and its associated microbiota.

Peptides have long been recognized as important signaling molecules in the hydra peptidome[24, 25, 38]. A variety of neuropeptides have been identified with different functions within the nervous system and in the dynamic interplay between neurons and other cell types. For example, Hym-357 strongly induces both tentacle and body contraction of normal polyps but has no effect on epithelial polyps, indicating that the peptide does not work directly on muscles but presumably activates other neurons, which in turn release neurotransmitters to directly induce muscle contraction[22, 23, 25]. Many of these peptides are encoded by taxonomically restricted genes[39]. Indeed, peptides of the NDA-1 family are specific for the genus *Hydra* and are absent in other animal taxa. However, we speculate that the function of neuropeptides as regulators of a resident microbiome may be widespread in the animal kingdom.

It should be noted that the tissue covered by the microbiome is heterogeneous. Neurons are in close proximity to epithelial cells, while the secretome creates a specific milieu to harbor particular bacterial colonizers. Neuropeptides can be considered part of the secretome that influence the microbial community composition by selective antimicrobial activities. The local distribution of neurons expressing specific neuropeptides contributes to local tissue heterogeneity and associated variation in microbiome composition.

The holobiont concept implies that all components participate in establishing resilience. Cnidarians have evolved a nerve net where sensory and ganglion neurons and their processes are interspersed among the epithelial cells of both layers. We speculate that the evolutionary process that resulted in a new organ such as the nervous system was partly driven by the necessity to maintain the holobiont. The observed selective effects of neuropeptides on commensal microbes not only contribute to solve the long-standing question of how community membership in a given holobiont is maintained, but also extends the functional capacity of the nervous system in early branching metazoans: their neuron-derived antimicrobial peptides may help to establish and maintain the species-specific microbiome in a spatially controlled manner. Recognizing the relationship between the neuroimmune system and the associated microbiome may provide novel insights into a deeper understanding and improved management of a number of disorders in which the neuron-microbe dialogue is disturbed.

## Methods

**Chemicals.** If not other stated otherwise, chemicals were obtained from Roth, Karlsruhe, Germany. All primers were obtained from MWG Eurofins, Ebersberg, Germany.

**Animals used and culture conditions.** Experiments were carried out using *Hydra vulgaris* AEP[40]. All laboratory-cultured strains of *H. vulgaris* AEP are available from the University of Kiel. Standardized culture medium, food (1st instar larvae of *Artemia salina*, fed 3× per week) and culture conditions are described elsewhere[41].

**Identification of NDA-1 peptide and structural modeling.** A fold recognition algorithm was validated with human CYP17A1 and cytochrome P450 (ProHit package, ProCeryon Biosciences GmbH, Salzburg, Austria). The X-ray structure of pancreatic colipase (PDB accession code 1N8S) produced a high score of the pair potential and the best match of cysteine-pairing; thus, this served as the template for the three-dimensional model of NDA-1. According to the alignment obtained by the fold recognition procedure, amino-acid residues were exchanged in the

template. Insertions and deletions in NDA-1 were modeled using a database-search approach included in the software package WHATIF[42].

**In situ hybridization and immunostaining of NDA-1.** In situ nucleotide hybridization in hydra polyps whole-mounts was performed as previously described[43]. A digoxigenin (DIG)-labeled antisense RNA probe was designed (primers NDA-1F: 5′-AACAAAATCAATTTCGCTGA-3′ and NDA-1R: 5′-TCGTCATTTA AATCATCTTC-3′) to recognize specifically the sequence of *Hydra NDA-1* gene product (GenBank accession number XM_002162825). DIG-labeled sense probes (targeting the same sequences as the antisense probes) were used as a control.

For Immunocytochemistry, polyclonal antibodies were raised against the C-terminal part of NDA-1 in rabbits. The synthetic peptide ((C)TNEDDLNDER KYFKQDK) (Genosphere Biotechnologies, Paris, France) was coupled to KLH prior to injection. The resulting polyclonal serum was affinity-purified and used for immunocytochemistry. Detection of NDA-1 protein in whole mounts was performed following standard procedures[44]. In brief, following anesthesia, polyps were fixed in 4% (v/v) paraformaldehyde for 30 min. After removal of fixative and permeabilization with 0.5% Triton X-100 in PBS, polyps were incubated in blocking solution containing 1% bovine serum albumin (BSA). NDA-1-antiserum (1:160 dilution, 5.6 ng/µL) in blocking buffer was incubated at 4 °C overnight. After four washing steps with blocking solution for 10 min each, immunostaining with NDA-1-specific rabbit serum was followed by secondary goat anti-rabbit 488 IgG antibodies (Molecular Probes, Invitrogen, Eugene, OR, USA, cat# A-11034) as described previously[45]. F-actin was stained by rhodamin-phalloidin and nuclear DNA by TO-PRO3 iodide (642/661; Molecular Probes, Invitrogen, Eugene, OR, USA, cat# T3605) as described previously[45]. The outer membrane surface of the ectodermal epithelial cells was stained by 1A10 monoclonal antibodies[46]. Confocal laser-scanning microscopy was performed using a TCS SP1 laser-scanning confocal microscope (Leica).

**Recombinant expression in *E. coli*, refolding and purification of rNDA-1.** A nucleotide fragment corresponding to amino-acid residues 21 to 92 of *NDA-1* was cloned into pet28a vector (Merck-Millipore/Novagen, Darmstadt, Germany) and expressed in *E. coli* BL21 as inclusion bodies. Recombinant bacteria were lysed at 0 °C with 50 mM Tris-HCl, 0.1 M NaCl, 5 mM EDTA, 0.5% (v/v) Triton-X100, 1 mM dithiothreitol (DTT), pH 8.0 using interval sonication (5 s impulse, 5 s pause) for 3 min. The suspension was centrifuged at 20,000×g for 10 min and after three repeated sonication/centrifugation steps the pellet (containing NDA-1) was washed with lysis buffer without Triton X-100 and DTT and stored at −20 °C. The denatured rNDA-1 was refolded by rapid dilution as follows. The inclusion body pellet was solubilized at 0 °C in solubilization buffer (8.5 M urea, 0.1 M Tris-base, 50 mM glycine, 0.1 M DTT, pH 8.0) by interval sonication for 1 min (5 s impulse, 5 s pause). Protein was concentrated by 20,000×g centrifugation and drop-wise diluted to a final concentration of 100 µg/mL (as estimated by A280 absorption) into refolding buffer (0.1 M Tris-base, 50 mM glycine, 10% (v/v) glycerin, 0.4 mM L-arginine, 5 mM glutathione red, 0.5 mM glutathione oxi, 1 mM EDTA, 0.2 mM PMSF, pH8.0) under moderate stirring at room temperature. The solution was stirred overnight at room temperature in the dark after which it was dialyzed (MWCO 3500, ThermoFisher Scientific (Pierce), Waltham, MA, USA) at 4 °C against 0.1 M Tris-base, 0.15 M NaCl, pH 8.0. Any precipitated protein was removed by centrifugation at 20,000×g for 30 min at 4 °C. The supernatant containing water-soluble rNDA-1 peptide was again dialyzed overnight at 4 °C and finally stored at 4 °C. The dialysate was further purified with two consecutive reverse phase chromatography procedures. To decrease the volume and as a pre-purification step, the dialysate was bound to a SepPak® C18 Vac 6 cc column (Waters, Eschborn, Germany) and eluted with 84% acetonitrile (ACN). The eluate was lyophilized and stored at −20 °C. The dried rNDA-1 peptide was solubilized in 0.05% trifluoroacetic acid (TFA) and bound to an Xselect® CSH C18 reverse phase HPLC column (Waters, Eschborn, Germany) after which a linear gradient of 1.2% ACN/min for 60 min was used for elution. The peptide-containing fractions were lyophilized and solubilized in 0.05% (v/v) TFA and further analyzed for their antibacterial activity using *Bacillus megaterium* ATCC 14581 (source: American Type Culture Collection (ATCC)) in a microdilution susceptibility assay. Fractions showing a MIC < 0.46 µM were pooled and analyzed by sodium dodecyl sulfate-polyacrylamide gel electrophoresis, mass spectrometry and circular dichroism spectroscopy. The pooled fractions were aliquoted and stored at −20 °C to be further used in a microdilution susceptibility assay.

**MIC determination of antibacterial neuropeptides.** The following bacterial strains were used in MIC assays: *Bacillus megaterium* ATCC 14581, *Staphylococcus aureus* ATCC 12600 (both obtained from ATCC), *Bacillus subtilis* DSM 10, *Trichococcus pasteurii* DSM 2381, *Trichococcus collinsii* DSM 14526 (all three obtained from Deutsche Stammsammlung von Mikroorganismen und Zellkulturen GmbH (DSM-Braunschweig), Germany), *Escherichia coli* K12 MG1665 (donated by Dr J.C. Escalante-Semerena, MMI UW-Madison, WI, USA), and three isolates from hydra cultures: *Curvibacter* sp[16], *Acinetobacter* sp., and *Pseudomonas* sp. The latter two strains were isolated from biofilms formed in hydra culture dishes (S. Fraune, unpublished data).

Microdilution susceptibility assays were performed in 96-well microtiter plates that were pre-coated with sterile 0.1% BSA for 10 min. After removal of BSA the wells were filled with a twofold dilution series of either rNDA-1, Hym-370, Hym-357, or RFamid III peptide. The neuropeptides Hym-370 (KPNAYKGKLPI GLW-amide), Hym-357 (KPAFLFKGYKP-amide) and RFamid III (KPHLRGRF-amide) had been chemically synthesized (GenScript, Hong Kong, China) at quantities between 5–9 mg with a purity of > 95% and lyophilized peptides were dissolved in 0.05% TFA to final concentration of 10 mg/mL. Incubation with an inoculum of approximately 100 CFU per well was performed in 10 mM Na₂HPO4 buffer (pH 6.2), except for the three strains derived from hydra cultures, which were incubated in R2A media. Following overnight incubation at 37 °C (*B. megaterium*, *S. aureus*, *B. subtilis*, *E. coli*) or for 3–5 days at 18 °C (others) in a moisture chamber the MIC was determined as the lowest serial dilution showing absence of a bacterial cell pellet. Experiments were carried out in triplicates and MICs were reported as a range in case of triplicate variation.

**Generation of transgenic *Hydra vulgaris* AEP polyps.** Hairpin-mediated silencing of target genes in hydra was achieved as previously described[47]. For generating NDA-1 knockdown lines, a cassette consisting of a 225-bp-long fragment of *NDA-1* and its corresponding antisense sequence, separated by a spacer of 157 bp, was cloned in the LigAF-1 vector downstream of *eGFP*. The construct was injected into *H. vulgaris* AEP embryos as previously described[40]. Founder polyps showing stable eGFP expression in a group of interstitial stem cells (i-cells) were expanded further by clonal propagation. By selecting for eGFP-expression, mass cultures of polyps without transgenic i-cells (NDA-1 control) as well as knockdown polyps expressing eGFP (NDA-1 kd) were generated. Completely transgenic NDA-1 knockdown lines could not be established, due to the low eGFP expression (a hallmark of the hairpin construct), the sophisticated structure of the nerve net with countless numbers of i-cells, and the fact that transgenic hydra start as mosaic animals with only a few transgenic cells. As a result, NDA-1 expression in the isolated lines can be variable. Thus, *NDA-1* knockdown was validated by quantitative real-time PCR using specific oligonucleotide primers (AL8F: 5′-GAGCAAGCGTTTGCATGAGC and AL8R: 5′-CTTCATCGGGTTCAGGTTCG). Reverse transcription PCR (RT-PCR) data were analyzed by conventional ΔΔCt analysis, using expression levels of hydra Elongation Factor 1α (EF1αF: 5′-GCAGTACTGGTGAGTTTGAAG and EF1αR: 5′-CTTCGCTGTATGGTGGTTCAG) and hydra β-actin (hyActinF: 5′-GAATCA GCTGGTATCCATGAAAC and hyActinR: 5′-AACATTGTCGTACCACCTGA TAG) as equilibration references. Fold-changes were normalized to control polyps of the same line (D1, E11).

To analyze the effect of *NDA-1* knockdown on abundance of *Curvibacter* sp. at different hydra body regions, the knockdown lineages D1 and E11 were each cultured with their controls as a mixed animal population. The polyps were starved for 2 days and GFP-positive knockdown animals were separated from GFP-negative control animals prior to *Curvibacter* analysis.

**Spatial distribution of *Curvibacter* sp. on hydra polyps.** In three to five independent experiments, combined hydra and bacterial DNA was isolated from ten polyps each that were first washed three times with sterile filtered culture medium and then divided into five body sections: tentacles, head (hypostome area without tentacles), upper body half, lower body half, and foot (peduncle). These tissue parts were subjected to the DNeasy Blood and Tissue Kit (Qiagen, Hilden, Germany) following the manufacturer's protocol to extract DNA, except that in the final step DNA was eluted in 50 µL. The amount of isolated DNA from the various body parts was standardized using quantitative reverse transcription PCR (qRT-PCR) targeting β-actin as described above. Total bacterial DNA was quantified using universal bacterial 16 S rDNA primers (Eub338F: 5′-ACTCCTACGGGAGGCAG CAG; and Eub518R: 5′-ATTACCGCGGCTGCTGG) and *Curvibacter* sp. DNA was quantified using genus-specific primers (CP_AEP1.3 F: 5′-TAGCGAGCTCTA ATACAGTTTGCTA and CP_AEP1.3 R: 5′-GGGATTTCACATCTGTCTTAC ATC). The latter experiments were performed as threefold (Fig. 3d, e) or fivefold (Fig. 3a) replicates.

**Statistical analysis.** Statistical analyses were performed using R software (version 3.2.3). For Fig. 3a (influence of tissue on total bacteria or *Curvibacter* sp. abundance), data were analyzed separately by generalized linear model with a gamma distribution (GLM; $n = 5$). The explanatory variable was *tissue*. The method of contrasts was used for side-by-side comparisons. P-values were adjusted for multiple comparisons using the false discovery rate's correction (fdr). For Fig. 3d (effect of knock-down on the expression of NDA-1) fold-change data were log transformed and a linear model with two factors (*treatment* and *line*) was used. For analysis of variance (ANOVA, $n = 3$), residuals were checked for normality by Shapiro's test, and for homoscedasticity by Levene's test. F statistics were given for global effects of factors and pairwise comparisons were performed using Tukey's honestly significant difference (HSD) test. For Fig. 3e (influence of treatment on bacterial abundance in the different tissues), data ($n = 3$) were analyzed by generalized linear model with a gamma distribution. The minimum model that fitted the data best included *tissue* and *treatment* as explanatory variables (Supplementary Table 1). $\chi^2$ statistics were given for global effects of factors and the method of contrasts was used for side-by-side comparisons. P-values were adjusted for multiple comparisons using the fdr correction.

**Data availability**. The authors declare that the data supporting the findings of the study are available in this article and its Supporting Information files, or from the corresponding author upon request.

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

## Acknowledgements

We thank Natacha Kremer for help with the statistical analysis, Alexander Klimovich for assistance in confocal microscopy and Matthias Leippe for providing the HPLC facility. This work was supported by the Deutsche Forschungsgemeinschaft (DFG) (CRC1182 "Origin and function of Metaorganisms", DFG grant BO 848/17–1, and grants from the DFG Cluster of Excellence program "Inflammation at Interfaces"). T.C.G.B. gratefully appreciates support from the Canadian Institute for Advanced Research (CIFAR).

## Author contributions

R.A. and K.S. were responsible for the experimental design and contributed equally to prepare the manuscript. R.A. performed protein purification, MIC assays, qRT-PCR, and (with K.S.) immunocytochemistry. K.S. generated the constructs for transgenesis and recombinant expression in *E. coli* (with E.-M.H.), performed in situ hybridization and mechanical stimulation. A.P.M.R. performed qRT-PCR and the GSH assay. F.A.-E. was responsible for confocal microscopy. M.S. performed the phylogenetic analysis. J.G. was

responsible for the structural modeling of NDA-1. T.M.W. assisted in graphical presentation of the data and writing the manuscript. T.G.C.B. supervised the project and was the driving force behind the research concept.

## Additional information

**Competing interests:** The authors declare no competing financial interests.

