## [Peer Review File · Nature Communications]

Reviewers' comments:

Reviewer #1 (Remarks to the Author):

Review of Augustin et al 2017 nature communications: The neuronal secretome configures a host-specific microbiome.

This exciting paper by Augustin and colleagues makes a major contribution to our understanding of the exquisite interplay between hosts and their microbiomes. Through a series of elegant experiments the Authors determined that sensory and ganglion neurons in the hydra epithelium secrete neuropeptides with antimicrobial activity and that this activity plays a fundamental role in structuring the microbiome. The experimental work to validate their hypothesis was comprehensive and the Authors should be congratulated on testing additional hydra neuropeptides to better understand the ubiquity of this phenomenon. The microscopy work is outstanding and this is a beautiful study.

Despite my enthusiasm for this paper, a number of aspects are treated rather superficially, particularly in the discussion:

- 1) From the abstract I was expecting the ms to focus on the disappearance of the main Gram +ve bacteria over developmental time due to an increase in NDA-1 secretion, however the Authors focus almost exclusively on the distribution and correlation of the *Curvibacter* coloniser with NDA-1. Some discussion should be made of the removal of the Gram +ve symbionts.
- 2) The discussion provides limited interpretation of why the hydra would need this microbial transition / restructuring? This may be covered in other manuscripts but an expansion of the discussion to include basic information on the microbiome would be valuable. For instance, how does this hydra actually acquire its microbiome? Is it vertically transmitted? If so, what would be the value for the holobiont investing in vertical transmission, only to replace the transmitted symbionts within the 1st weeks of life? Do we know the functional role of the major symbionts in this species of Hydra? If so, this should be briefly summarised to provide holobiont context and establish the importance of the microbiome.
- 3) Fig 1b shows considerable variation in the Gram +ve bacterium over time yet this is not discussed at all?
- 4) Line 97- expand the AEP reference?
- 5) The main text (lns 50-52) seems to imply that the Authors will show the evolution of these neuropeptides but this is not developed in the manuscript, outside lines 97-100 which states that there are no identifiable orthologs to NDA-1 outside of Hydra?
- 6) The colouring of the hydra body in Fig 3B is somewhat incongruent with the text and Fig 3A as it shows a white foot, but the head appears to be the same brown as the rest of the body.
- 7) Why is the y axes on Fig 3D broken into 2 sections, this seems unnecessary as 2 separate scales are not actually needed. Whilst the expression change in the knockdowns (3C) is highly convincing, the variance in the related *Curvibacter* abundance (3D) is very high and less convincing, although I appreciate that the Authors have statistical significance.
- 8) In the experimental methods (lines 495) please clarify whether you used n=10 hydra for each of the 3 independent experiments or in total?

Reviewer #2 (Remarks to the Author):

Interactions of microbiota and its host play important roles in host's health and disease. The change in the interactions appears responsible for maintaining homeostasis and development of a particular disease. It is also known that the components of microbiota changes depending on tissues, organs or whole organisms, or developmental stages. Myriad of molecules must be involved in the interactions. Last ten years, neuropeptides with antimicrobial activity are shown to be a part of such molecules. The authors' group have identified species- and developmental stage-specific microbiota in Hydra, and in this article tried to identify such molecules. The authors

identified a novel molecule called NDA-1, which turns out to be neuropeptides. From the expression pattern of NDA-1 gene and specific inhibitory effect on various bacterial species, the authors proposed that NDA-1 regulates the composition of the microbiota. Furthermore, The authors claim that NDA-1 determines the spatial distribution of hydra's main colonizer, *Curvibacter* sp, by observing negative correlation between abundance of *Curvibacter* and the presence of NDA-1 expressing neurons along the body axis.

Major comment: However, Fig 3 does not show the negative correlation. Why *Curvibacter* in the head where NDA-1 expressing neurons are present is more abundant in the body where NDA-1 expressing neurons are fewer? Fig. 3 suggests that some other factors must be involved in repressing *Curvibacter*. To claim that NDA-1 determines the spatial distribution of hydra's main colonizer, *Curvibacter* sp, it is essential to identify additional factor(s).

The authors have identified several other antimicrobial peptides and in this article showed that three known neuropeptides also have antimicrobial activity against different repertoire of bacteria, which is very interesting. However, it is also possible to find more antimicrobial peptides in Hydra. Thus, as originally believed, it seems far more complicated to maintain a particular composition of microbiota at a given condition.

Minor commentx:

Line 107-108 (Fig.2D-F): From the resolution of the Figure it is not possible to tell the expression of NDA-1 mRNA in ganglion and/or sensory neurons.

Fig. 2I,J show explicitly the expression in both ganglion and sensory neurons.

Fig.3D. Did authors check the abundance of *Curvibacter* in the tentacles of knockdown animals?

Reviewer #3 (Remarks to the Author):

This is an interesting paper reporting the identification, expression and functional analysis of a novel neuropeptide, NDA-1 from Hydra. The authors are able to show that this neuropeptide has antimicrobial activity and they postulate that the secretion of NDA-2 helps shaping the microbial community of hydra. They also investigate the role of three other, previously published neuropeptides, RFa III, Hym-370 and Hym-357 and show that they also have antimicrobial activity in MIC assays. An antimicrobial role for a neuropeptide would lend support to a notion put forward by Kim Brodgen and others that neuropeptides often are involved in innate immunity in vertebrates and I therefore think that this paper will be an important contribution to this discussion.

The data are for the most part convincing. However, before publication, two major issues have to be addressed:

1) By MIC assays (please explain to the uninitiated reader) the authors show that very low concentrations (0.4 μ M) of NDA-1 can suppress growth of *Curvibacter*, but also other gram-positive or gram-negative bacteria. They also knockdown NDA-1 in two independent transgenic lines expressing shRNA against NDA-1, showing that it increases the concentration of *Curvibacter* specifically in the regions where the NDA-1 neurons are located (hypostome, foot). However, since *Curvibacter* is by far the most abundant microbe in Hydra, it is somewhat counter-intuitive that even very low concentrations of NDA-1 appear to have only modest effects on shaping the distribution of *Curvibacter*. Its graded distribution (highest in tentacles lowest in the foot) also does not really follow the rather two-parted location of NDA-1 neurons. Furthermore, the authors state that *Curvibacter* is mostly associated to the mucus, yet the NDA-1 neurons of the foot are ganglionic neurons, which have a basiepithelial location. It is unclear, how this affects the distribution of the secreted peptides in this region, as the antibody appears to detect the peptide only within the cells.

While the knockdown data show an increase of *Curvibacter* in head and foot, but not in the body column, the effects are barely significant. To independently confirm the antimicrobial effect of NDA-1, a gain-of function experiment would be desirable and to my knowledge also doable in Hydra: the authors should use transgenics to overexpress NDA-1 in all cells or at least all ectodermal epithelial cells under a ubiquitous promoter. This should drastically diminish the levels of *Curvibacter* in the animal. It would thereby also show, whether a changed bacterial community has any biological relevance for the organism.

2) The knockdown of NDA-1 should also be used to test for other functions, e.g. motion control, sensitivity to mechanical or chemical cues.

Minor points:

(Unfortunately neither line numbers nor page numbers are given.)

1) Table 1 and Fig. 4B should be combined in one table to make the values for NDA-1 comparable to the other peptides.

2) Fig. 3D: what does the Y-axis mean (*Curvibacter* sp. abundance relative to body)?

3) Fig. 4A. One would like to see the original stainings of Hym-370, Hym-357, and RFamide III, at least in a supplemental figure, not only the schematic drawing.

Fig. 4A. The schematic drawing would be clearer, if each neuropeptide was shown individually.

4) in the discussion the authors state: "supporting the view that from the very beginning of multicellular life, nerve cells have been involved in maintenance of the animal holobiont." This is an overstatement, as Hydra is a representative of a sister group of Bilateria, but certainly not the earliest off-shoot of multicellular life. So far, there is no information about the microbiome of ctenophores, sponges and placozoans (although it is likely to exist). Yet, sponges and placozoans do not possess neurons, hence, it is not clear, whether this antimicrobial peptides produced by neurons is an ancestral trait. Moreover, it seems that these peptides are rather taxonomically restricted genes, hence they could have evolved only in the lineage leading to hydra.

5) Likewise, the statement " Indeed, peptides of the NDA-1 family are specific for the genus hydra and are absent in other animal taxa. However, their function as regulators of a resident microbiome is probably universal in the animal kingdom." is contradictory in itself. How can NDA-1 have a universal role in the animal kingdom if it is only found in hydra? Please rephrase.

6) The MIC assay shows that NDA-1 has quite distinct effects on both specific Gram-positive and Gram-negative bacteria. The authors should briefly discuss the possible mechanisms for this specificity.

7) while the overall level of english is decent, there are some spelling and semantic errors. Please have it read by a native speaker.

(e.g. amphypathicity should read amphipathicity; awaits should read wait, etc.)

Below we renumbered and discuss the referee's comments in detail. Our responses and changes to the manuscript are given in *italics*.

Reviewer #1 (Remarks to the Author):

Review of Augustin et al 2017 nature communications: The neuronal secretome configures a host-specific microbiome.

This exciting paper by Augustin and colleagues makes a major contribution to our understanding of the exquisite interplay between hosts and their microbiomes. Through a series of elegant experiments the Authors determined that sensory and ganglion neurons in the hydra epithelium secrete neuropeptides with antimicrobial activity and that this activity plays a fundamental role in structuring the microbiome. The experimental work to validate their hypothesis was comprehensive and the Authors should be congratulated on testing additional hydra neuropeptides to better understand the ubiquity of this phenomenon. The microscopy work is outstanding and this is a beautiful study.

Despite my enthusiasm for this paper, a number of aspects are treated rather superficially, particularly in the discussion:

1. From the abstract I was expecting the ms to focus on the disappearance of the main Gram +ve bacteria over developmental time due to an increase in NDA-1 secretion, however the Authors focus almost exclusively on the distribution and correlation of the *Curvibacter* coloniser with NDA-1. Some discussion should be made of the removal of the Gram +ve symbionts.

Thank you for this comment and suggestions. We appreciate the advice and now have added a sentence to the discussion (line 246) summarizing the effects of NDA-1 on both Gram-positive and Gram-negative bacteria: "Using both in vivo transgenesis and in vitro antimicrobial activity assays we have demonstrated that neuropeptides influence the hydra microbiome by inhibiting the growth of Gram-positive bacteria and by shaping the spatial localization of the main colonizer, the Gram-negative bacterium Curvibacter sp."

2. The discussion provides limited interpretation of why the hydra would need this microbial transition / restructuring? This may be covered in other manuscripts but an expansion of the discussion to include basic information on the microbiome would be valuable. For instance, how does this hydra actually acquire its microbiome? Is it vertically transmitted? If so, what would be the value for the holobiont investing in vertical transmission, only to replace the transmitted symbionts within the 1st weeks of life? Do we know the functional role of the major symbionts in this species of

Hydra? If so, this should be briefly summarised to provide holobiont context and establish the importance of the microbiome.

Thank you for the advice to be more explicit about the microbiome in Hydra. We now have extended the discussion by adding a short paragraph (line 253-258) which explains how the microbiome is obtained: "Considering Hydra's life history, microbial colonization in early embryos is controlled by maternally encoded antimicrobial peptides (Fraune et al., 2010). After mid-blastula transition, zygotically expressed antimicrobial peptides take control of the microbiome. After hatching, a stable microbiome is established within 3 to 4 weeks (Franzenburg et al., 2013). Symbiotic microbes may be acquired by both vertical and horizontal transmission (Fraune et al., 2010). In adult hydra, interaction of stably associated microbes plays a role in pathogen defense (Fraune et al., 2014)."

3. Fig 1b shows considerable variation in the Gram +ve bacterium over time yet this is not discussed at all?

This is a good point and we have now added a sentence to the text (lines 88-90) stating: "The high variation observed between weeks 3 and 6 was shown previously (Franzenburg et al., 2013) to be a characteristic feature of the maturation of the adult microbiome in hydra."

4. Line 97- expand the AEP reference?

We have added the reference as suggested.

5. The main text (lines 50-52) seems to imply that the Authors will show the evolution of these neuropeptides but this is not developed in the manuscript, outside lines 97-100 which states that there are no identifiable orthologs to NDA-1 outside of Hydra?

It seems our text was confusing as this is a misunderstanding. We have removed the misleading sentence concerning the evolutionary origin and now refer to Supplementary Figure S1 (line 102), which shows that NDA-1 is a taxonomically restricted gene.

6. The colouring of the hydra body in Fig 3B is somewhat incongruent with the text and Fig 3A as it shows a white foot, but the head appears to be the same brown as the rest of the body.

Thank you for your comment, which was also made by referee 3. We fully agree that this figure was confusing and did not make the role of NDA-1 expressing neurons in controlling the Curvibacter colonization along the body column clear enough. The difficulty in understanding might arise from the fact that Curvibacter for unknown reasons is extremely abundant in tentacles. The core finding

presented in Figure 3C (former 3B) is that the presence of NDA-1 neurons causes an average 6-fold reduction of Curvibacter abundance compared to tentacles; and an average 2-fold reduction of Curvibacter in foot tissue compared to the moderate abundance of Curvibacter in body column tissue.

We have revised Figure 3 to clarify that NDA-1 expressing neurons are responsible for a drastic change in Curvibacter abundance between tentacle vs head and body column vs foot tissue. We have added an additional panel (Fig. 3B) showing the direct comparison of Curvibacter abundance in the two adjacent tissues tentacle/head and body column/foot.

7. Why is the y axes on Fig 3D broken into 2 sections, this seems unnecessary as 2 separate scales are not actually needed. Whilst the expression change in the knockdowns (3C) is highly convincing, the variance in the related Curvibacter abundance (3D) is very high and less convincing, although I appreciate that the Authors have statistical significance.

In principle we fully agree with the referee. However, in reply to referee #2 we now have added data showing the abundance of Curvibacter in tentacle tissue. This makes it necessary to break the y-axes in 2 sections.

8. In the experimental methods (lines 495) please clarify whether you used n=10 hydra for each of the 3 independent experiments or in total?

We used 10 hydra polyps for each independent experiment; this is now clarified in line 534.

Reviewer #2 (Remarks to the Author):

Interactions of microbiota and its host play important roles in host's health and disease. The change in the interactions appears responsible for maintaining homeostasis and development of a particular disease. It is also known that the components of microbiota changes depending on tissues, organs or whole organisms, or developmental stages. Myriad of molecules must be involved in the interactions. Last ten years, neuropeptides with antimicrobial activity are shown to be a part of such molecules. The authors' group have identified species- and developmental stage-specific microbiota in Hydra, and in this article tried to identify such molecules. The authors identified a novel molecule called NDA-1, which turns out to be neuropeptides. From the expression pattern of NDA-1 gene and specific inhibitory effect on various bacterial species, the authors proposed that NDA-1 regulates the composition of the microbiota. Furthermore, The authors claim that NDA-1 determines the spatial distribution of hydra's main colonizer, Curvibacter sp, by observing negative correlation between abundance of Curvibacter and the presence of NDA-1 expressing neurons along the body axis.

9. Major comment: However, Fig 3 does not show the negative correlation. Why *Curvibacter* in the head where NDA-1 expressing neurons are present is more abundant in the body where NDA-1 expressing neurons are fewer? Fig. 3 suggests that some other factors must be involved in repressing *Curvibacter*. To claim that NDA-1 determines the spatial distribution of hydra's main colonizer, *Curvibacter* sp, it is essential to identify additional factor(s).

The authors have identified several other antimicrobial peptides and in this article showed that three known neuropeptides also have antimicrobial activity against different repertoire of bacteria, which is very interesting. However, it is also possible to find more antimicrobial peptides in Hydra. Thus, as originally believed, it seems far more complicated to maintain a particular composition of microbiota at a given condition.

Thank you for this constructive criticism. In response to this comment, and that of the other reviewers, we have thoroughly revised Figure 3 and rewritten the corresponding part of the main text to clarify that NDA-1 neurons are responsible for the drastic change in Curvibacter abundance at the transition zone of tentacle vs head and body column vs foot tissue. The additional panel 3B now clearly indicates the differences of Curvibacter abundance in the two adjacent tissues tentacle/head and body column/foot. The results show a clear correlation between the presence of NDA-1 expressing neurons and Curvibacter abundance in both body regions head and foot.

10. Minor comment:

Line 107-108 (Fig.2D-F): From the resolution of the Figure it is not possible to tell the expression of NDA-1 mRNA in ganglion and/or sensory neurons.

Fig. 2I,J show explicitly the expression in both ganglion and sensory neurons.

The pictures in Figure 2D-F present in situ hybridizations with an NDA-1-specific probe, which visualizes the corresponding RNA in the cytosol. This does not allow identifying the specific neuronal cell type. Figure 2I and J present examples of immunocytochemistry using an NDA-1 specific antibody, which detects the peptide in the cell body and allows to unambiguously differentiate between ganglion and sensory neurons. Figure 2K has been modified outlining an NDA-1 positive sensory neuron with a broken white line.

11. Fig.3D. Did authors check the abundance of *Curvibacter* in the tentacles of knockdown animals?

We have now added data for the tentacles in new panel 3E. That figure shows clearly that there are no differences in Curvibacter abundance in tentacles between control and knockdown animals. This

supports our view that secreted NDA-1 peptide affects Curvibacter abundance in the transition zone between tentacles and head as well as body column and foot.

Reviewer #3 (Remarks to the Author):

This is an interesting paper reporting the identification, expression and functional analysis of a novel neuropeptide, NDA-1 from Hydra. The authors are able to show that this neuropeptide has antimicrobial activity and they postulate that the secretion of NDA-2 helps shaping the microbial community of hydra. They also investigate the role of three other, previously published neuropeptides, RFa III, Hym-370 and Hym-357 and show that they also have antimicrobial activity in MIC assays. An antimicrobial role for a neuropeptide would lend support to a notion put forward by Kim Brodgen and others that neuropeptides often are involved in innate immunity in vertebrates and I therefore think that this paper will be an important contribution to this discussion.

Thank you for this summary. We consider the Brodgen et al paper as a true landmark paper, which paved the way for the neuro-immuno field.

The data are for the most part convincing. However, before publication, two major issues have to be addressed:

12. By MIC assays (please explain to the uninitiated reader) the authors show that very low concentrations (0.4 μ M) of NDA-1 can suppress growth of Curvibacter, but also other gram-positive or gram-negative bacteria. They also knockdown NDA-1 in two independent transgenic lines expressing shRNA against NDA-1, showing that it increases the concentration of Curvibacter specifically in the regions where the NDA-1 neurons are located (hypostome, foot). However, since Curvibacter is by far the most abundant microbe in Hydra, it is somewhat counter-intuitive that even very low concentrations of NDA-1 appear to have only modest effects on shaping the distribution of Curvibacter.

Thank you for pointing this out. We have explained the abbreviation "MIC" in the legend of Table 1 and in the text (line 122). (A detailed description of the method is provided in material and method section). With regard to the high in vitro activity of NDA-1 and the "moderate" activity in vivo, it is important to consider that in vivo NDA-1 faces Curvibacter in the outer mucus layer of a complex glycocalyx. When secreted into the mucus layer there may be many factors interfering with NDA-1 activity including local pH, in vivo metabolic performance of Curvibacter and also factors like secreted proteases by the bacterial community. Nevertheless, we consider in vivo activity of NDA-1

as sufficient and comparable to in vitro, because it reduces Curvibacter abundance by on average 6-fold from tentacle to head as well as on average 2-fold from body column to foot (Fig. 3B).

13. Its graded distribution (highest in tentacles lowest in the foot) also does not really follow the rather two-parted location of NDA-1 neurons.

Thank you for your comment. As written in our reply to the other reviewers, the core finding presented in Figure 3 is that the NDA-1 neurons cause a significant reduction of Curvibacter abundance in head tissue compared to tentacle; and in foot tissue compared to the moderate abundance of Curvibacter in body column tissue. Also see our response to point 7 above.

14. Furthermore, the authors state that Curvibacter is mostly associated to the mucus, yet the NDA-1 neurons of the foot are ganglionic neurons, which have a basiepithelial location. It is unclear, how this affects the distribution of the secreted peptides in this region, as the antibody appears to detect the peptide only within the cells.

Thank you for your comment. We obviously had not made it clear enough that the NDA-1 expressing neuronal cell population in the foot also contains sensory neurons. A population of foot sensory neurons with protrusions that extend to the apical edge of the ectoderm has been previously described (Koizumi and Bode, 1991). Thus, similar to the head (Fig. 2I and J), neurons at the foot tissue are also sensory neurons able to target the colonizing microbiome in the mucus layer. We have added a broken white line to Figure 2K for clarity to outline a sensory neuron of the basal disc with protrusions reaching the mucus layer.

Koizumi, O. and H. R. Bode. 1991. "Plasticity in the Nervous System of Adult Hydra. III. Conversion of Neurons to Expression of a Vasopressin-like Immunoreactivity Depends on Axial Location." The Journal of Neuroscience: the official journal of the Society for Neuroscience 11(7):2011–20. Retrieved (<http://www.ncbi.nlm.nih.gov/pubmed/2066772>).

15. While the knockdown data show an increase of Curvibacter in head and foot, but not in the body column, the effects are barely significant. To independently confirm the antimicrobial effect of NDA-1, a gain-of function experiment would be desirable and to my knowledge also doable in Hydra: the authors should use transgenics to overexpress NDA-1 in all cells or at least all ectodermal epithelial cells under a ubiquitous promoter. This should drastically diminish the levels of Curvibacter in the animal. It would thereby also show, whether a changed bacterial community has any biological relevance for the organism.

This is a very good point. To examine the effect of NDA-1 in transgenic polyps ectopically expressing NDA-1 peptide in epithelial cells, we attempted to generate several transgenic lines with a constitutively active promoter expressing NDA-1:eGFP. As a control, polyps were injected with the same eGFP-construct lacking the NDA-1 sequence. The results are presented in a new Supplementary Figure S5. Although the NDA-1:eGFP construct worked properly to get NDA-1:eGFP secreted into vesicles of ectodermal epithelial cells (Suppl. Fig. S5D), establishment of stable transgenic lines failed. Whereas the total hatching rate of embryos injected with the NDA-1:eGFP construct (n = 137) was comparable to the control construct (Suppl. Fig. S5E), we obtained only a few hatchlings expressing NDA-1:eGFP (Suppl. Fig. S5F). Furthermore, 70% of the NDA-1:eGFP expressing hatchlings failed to develop normally and eventually died (Suppl. Fig. S5G). Notably, the few surviving polyps lost all of their transgenic cells within 2-3 weeks after hatching. Thus, overexpression of NDA-1 results in a lethal phenotype for transgenic epithelial cells, indicating a disadvantageous interaction of NDA-1 with as yet unknown epithelial cell-specific factors. Because this is an interesting novel finding, we have added a corresponding sentence to the text (lines 205-207) and show the data in a new Supplementary Fig. S5.

16. The knockdown of NDA-1 should also be used to test for other functions, e.g. motion control, sensitivity to mechanical or chemical cues.

We agree that this is an interesting aspect. We decided, therefore, to perform three additional functional assays using both NDA-1 knockdown lines.

a) *In one new set of experiments we used the NDA-1 knockdown lines to test the activity of NDA-1 on motion control in an assay we have recently established to determine body column contraction frequency (Andrea P. Murillo Rincón et al., in prep.). For confidential perusal by the reviewer, we show here the spontaneous contraction frequency of NDA-1 knockdown and control polyps.*

Hydra polyps periodically contract in the absence of external stimuli. To quantify this behavior, we counted the number of spontaneous contractions per hour for individual, undisrupted polyps using time-lapse videos. After 30 min of acclimation the contraction frequency was measured for 60 min.

*We assessed the contraction frequency of the two transgenic NDA-1 knockdown lines (D1 and E11) displaying abnormal *Curvibacter sp.* abundance. When compared with their respective controls, we found no difference between the contraction frequency of the NDA-1 knockdown and the control lines. This observations indicate that NDA-1 is not involved in control of body column contractions.*

Since this behavioral assay is a major part of another paper describing the interaction between microbial colonizers and Hydra's nervous system (Andrea P. Murillo Rincón et al.,

in preparation), we decided to show these data for confidential perusal by the reviewer only , but decided not to include these in the current manuscript.

Spontaneous contraction frequency of NDA-1 knockdown and control polyps.

Spontaneous contraction frequency of NDA-1 knockdown lines does not differ from the frequency displayed by control lines (t-test: transgenic line D1: $F= 0.23 (1,43)$, $p= 0.63$, knockdown $n=28$, control $n=17$; transgenic line E11: $F= 01.56 (1,24)$, $p= 0.22$, knockdown $n=14$, control $n=12$).

- b) Second, we tested the functionality of the neuronal circuits activated by chemical stimuli in NDA-1 knockdown animals using a γ -glutamyl-cysteinyl-glycine (GSH) assay. The feeding response of Hydra can be evoked chemically in the presence of micromolar concentrations of reduced glutathione (GSH), which induces the polyps to open their mouth. This is a neuron-dependent behaviour as nerve-free animals do not open the mouth. When exposed to $10\mu\text{M}$ GSH all NDA-1 knockdown animals opened their mouth and there was no statistical significant difference in time to mouth opening compared to the control animals (new Supplementary Fig. S3).
- c) Third, NDA-1 knockdown animals were tested for sensitivity to mechanical cues. When Hydra polyps are pinched with a pair of forceps, the animals response with rapid contraction of the body column. NDA-1 knockdown and control animals showed no difference in contractile behavior when mechanically stimulated (new Supplementary Fig. S4).

A corresponding sentence was added to the main text (line 181-185): “The NDA-1 knockdown animals showed no differences in mechanical-induced contractile behavior and feeding response when compared to the corresponding control lines indicating that NDA-1 knockdown has no functional consequences regarding common behavioral responses in Hydra”

Minor points:

(Unfortunately neither line numbers nor page numbers are given.)

We apologize for this, which seems to be due to the electronic submission system. Other reviewers apparently could see page numbers and line numbers.

17. Table 1 and Fig. 4B should be combined in one table to make the values for NDA-1 comparable to the other peptides.

Thank you for pointing this out. As suggested by the referee, we have now incorporated the data presented in Table 1 into Figure 4B. Nevertheless, we deliberately kept Table 1 in, as the discussion and presentation of these data in the beginning of the text is essential for a logical build-up of our case.

18. Fig. 3D: what does the Y-axis mean (Curvibacter sp. abundance relative to body)?

This was indeed a bit confusing. We now have changed the y-axis description and removed the confusing phrase "relative to body". Curvibacter abundance in other tissue was measured relative to the Curvibacter abundance in body tissue (fold change qRT-PCR).

19. Fig. 4A. One would like to see the original stainings of Hym-370, Hym-357, and RFamide III, at least in a supplemental figure, not only the schematic drawing.

Thank you for this comment. Since the original stainings for these classical Hydra neuropeptides can be found in the corresponding original papers, which we refer to in the text (line 220) and in the legend of Figure 4, we decided not to incorporate them in the present manuscript.

20. Fig. 4A. The schematic drawing would be clearer, if each neuropeptide was shown individually.

Excellent suggestion. We have made corresponding changes to Figure 4.

21. in the discussion the authors state: "supporting the view that from the very beginning of multicellular life, nerve cells have been involved in maintenance of the animal holobiont." This is an overstatement, as Hydra is a representative of a sister group of Bilateria, but certainly not the earliest off-shoot of multicellular life. So far, there is no information about the microbiome of ctenophores, sponges and placozoans (although it is likely to exist). Yet, sponges and placozoans do not possess neurons, hence, it is not clear, whether this antimicrobial peptides produced by neurons is an ancestral

trait. Moreover, it seems that these peptides are rather taxonomically restricted genes, hence they could have evolved only in the lineage leading to hydra.

Thank you. We have decided to delete “the very beginning of multicellular life”.

22. Likewise, the statement " Indeed, peptides of the NDA-1 family are specific for the genus hydra and are absent in other animal taxa. However, their function as regulators of a resident microbiome is probably universal in the animal kingdom." is contradictory in itself. How can NDA-1 have a universal role in the animal kingdom if it is only found in hydra? Please rephrase.

We agree that this was confusing. We have rewritten the corresponding sentence and replaced “their function” by “the function of neuropeptides” (line 276).

23. The MIC assay shows that NDA-1 has quite distinct effects on both specific Gram-positive and Gram-negative bacteria. The authors should briefly discuss the possible mechanisms for this specificity.

This is an interesting point. The general view is that most AMPs act directly on cell membranes. Thereby, electrostatic interactions between the cationic peptides and negatively charged components of the cell wall such as LPS in Gram-negative and teichoic acids in Gram-positive bacteria are supposed to modulate the attraction to the bacterial surface. As shown in Figure 2C NDA-1 peptide is predicted to form a hydrophobic pocket, which might allow insertion into the microbial membrane. However, since a detailed mechanistic understanding of how NDA-1 interferes with bacterial growth is beyond the scope of our current work, we decided not to include a purely speculative paragraph.

24. while the overall level of english is decent, there are some spelling and semantic errors. Please have it read by a native speaker.

(e.g. amphypathicity should read amphipathicity; awaits should read wait, etc.)

Thank you. We have cleaned up typos and phrases.

REVIEWERS' COMMENTS:

Reviewer #1 (Remarks to the Author):

This is a greatly improved manuscript as the Authors have gone to considerable effort to incorporate the recommendations of all 3 Reviewers. I am satisfied that this will be a significant contribution for Nature Communications and have only a few additional minor editorial suggestions:

Line 205 Rephrase the start of this paragraph to 'Microbial colonization in early hydra embryos is controlled by maternally encoded antimicrobial peptides (31).

Line 210 Delete 'are' prior to 'pathogen defence'.

Line 218 Instead of 'wait to be analyzed' change to 'are still to be assessed'

Reviewer #2 (Remarks to the Author):

The revised text is now clear to the point. However, to make it clearer, it is advisable to modify Fig.3C: Since NDA-1 expressing neurons appear to be present in the body column, although their density seems lower (Fig. 2D), it is advisable to show them in the illustration (also Fig.4A). Or mention in the text (page 4) that NDA-1 expressing neurons are present at a lower density in the body column.

Minor points:1. What is ns in Fig. 2K? 2. Line148-149 It should be Fig. D instead of Fig. 3C

Reviewer #3 (Remarks to the Author):

The authors have largely satisfactorily addressed my (and I believe the other reviewers) concerns and comments. I think this will be a very nice paper with many new unexpected findings and I predict it will be widely cited.

However, a few points remain to be solved:

Fig. 3D clearly shows that two independent shRNA lines knockdown significantly the expression of NDA-1, but there is basically no change of relative abundance of *Curvibacter* (Fig.3E). Then in Fig. 3D, why do the authors show the statistical significance between the two shRNA transgenic lines (control and knockdown) rather than comparing knockdown with control in each line. It also contradicts the text (line 156-162). Is it simply a mislabeling in the legend of the figure?

Fig. 3

Minor points:

Fig. 3B is dispensable, because these data are shown in virtually the same manner in Fig. 3A. The legend of the axis in Fig. 3D should be "fold change NDA-1 expression".

REVIEWERS' COMMENTS:

Reviewer #1 (Remarks to the Author):

This is a greatly improved manuscript as the Authors have gone to considerable effort to incorporate the recommendations of all 3 Reviewers. I am satisfied that this will be a significant contribution for Nature Communications and have only a few additional minor editorial suggestions:

- Line 205 Rephrase the start of this paragraph to 'Microbial colonization in early hydra embryos is controlled by maternally encoded antimicrobial peptides (31).
- Line 210 Delete 'are' prior to 'pathogen defence'.
- Line 218 Instead of 'wait to be analyzed' change to 'are still to be assessed'

ALL DONE AS SUGGESTED

Reviewer #2 (Remarks to the Author):

The revised text is now clear to the point. However, to make it clearer, it is advisable to modify Fig.3C: Since NDA-1 expressing neurons appear to be present in the body column, although their density seems lower (Fig. 2D), it is advisable to show them in the illustration (also Fig.4A). Or mention in the text (page 4) that NDA-1 expressing neurons are present at a lower density in the body column.

Thank you for pointing this out. We have added a sentence to the text to clarify the expression pattern of the NDA-1 gene (line 113-115).

Minor points:

1. What is ns in Fig. 2K?

We explain 'ns= not significant' in the figure legend now.

2. Line148-149 It should be Fig. D instead of Fig. 3C

DONE

Reviewer #3 (Remarks to the Author):

The authors have largely satisfactorily addressed my (and I believe the other reviewers) concerns and comments. I think this will be a very nice paper with many new unexpected findings and I predict it will be widely cited.

However, a few points remain to be solved:

Fig. 3D clearly shows that two independent shRNA lines knockdown significantly the expression of NDA-1, but there is basically no change of relative abundance of *Curvibacter* (Fig.3E). Then in Fig. 3D, why do the authors show the statistical significance between the two shRNA transgenic lines (control and knockdown) rather than comparing knockdown with control in each line. It also contradicts the text (line 156-162). Is it simply a mislabeling in the legend of the figure?

Thank you for this comment. Indeed there was a mislabeling in the legend of Figure 3E. we have corrected this so now the figure displays exactly what the reviewer asks for: Figure 3E

shows a significant change in Curvibacter abundance in the head and foot between control (black and grey bars) and knockdowns (checkered bars).

Fig. 3 Minor points:

Fig. 3B is dispensable, because these data are shown in virtually the same manner in Fig. 3A.

The additional panel Fig3B was added in the last revision in response to other reviewers to clarify the core finding presented in Figure 3A, which is that NDA-1 expressing neurons are responsible for a drastic change in Curvibacter abundance between tentacle vs head and body column vs foot tissue.